# Effects of Long-Term Supplementation of Bovine Colostrum on Iron Homeostasis, Oxidative Stress, and Inflammation in Female Athletes: A Placebo-Controlled Clinical Trial

**DOI:** 10.3390/nu15010186

**Published:** 2022-12-30

**Authors:** Mirosława Cieślicka, Joanna Ostapiuk-Karolczuk, Harpal S. Buttar, Hanna Dziewiecka, Anna Kasperska, Anna Skarpańska-Stejnborn

**Affiliations:** 1Department of Physiology, Collegium Medicum in Bydgoszcz, Nicolaus Copernicus University in Toruń, M. Sklodowskiej-Curie 9, 85-094 Bydgoszcz, Poland; 2Department of Biological Sciences, Faculty of Physical Culture in Gorzow Wielkopolski, Poznan University of Physical Education, Estkowskiego 13, 66-400 Gorzów Wielkopolski, Poland; 3Department of Pathology & Laboratory Medicine, Faculty of Medicine, University of Ottawa, Ottawa, ON K1H 8M5, Canada

**Keywords:** intense exercise, female athletes, bovine colostrum supplementation, antioxidant and anti-inflammation effects, iron homeostasis, placebo-controlled clinical trial

## Abstract

Bovine colostrum supplementation has been suggested as a potential factor in reducing oxidative stress and inflammation. The primary objective of this study was to evaluate the effects of six months of bovine colostrum supplement intake (3.2 g; four capsules/day) in highly trained female athletes on changes in oxidative stress level, inflammation, and iron metabolism biomarkers after intense exercise. In this study, 20 trained female athletes were recruited. Participants were divided into two groups: 11 in the bovine colostrum (6-month supplementation) and 9 in the placebo group (6-month placebo supplementation). All participants completed an intense exercise test at the beginning of the experiment and after six months post-treatment. Blood samples were taken before, following exercise, and after 3 h recovery. Compared to the placebo group, the colostrum group showed a significant decrease in TBARS level (*p*< 0.01) at all time points, whereas a marked increase was observed in IL-6 (*p* < 0.01; pre-exercise) and SOD activity (*p* < 0.01), and transferrin (*p* < 0.01; rest period) and lactoferrin (*p* < 0.05; post-exercise) levels. The results suggested that 6-months of bovine colostrum supplementation is beneficial in the reduction of the harmful effects produced by free radicals (ROS), oxidative stress, and inflammation. In consequence, alleviation of the inflammatory response by bovine colostrum supplementation may also cause positive action on iron homeostasis in female athletes.

## 1. Introduction

The research results published so far indicate that the increased metabolism determining the energy supply of physical effort is accompanied by an increased generation of reactive oxygen species (ROS), which may also cause disturbances in the proper functioning of the hemopoietic system in the body [1,2,3]. The reduction of oxidative stress is especially essential for high-intensity or long-duration physical activity. However, the mechanism of this process is not fully understood. It is assumed that excessive production of ROS, among other factors, leads to the damage of erythrocyte membranes (as a consequence of lipid peroxidation), thus increasing their sensitivity to disintegration. As a result of increased hemolysis of erythrocytes, there is a significant increase in redox-active iron in circulation. The Fenton reaction ((Fe^2+^ + H_2_O_2_ →Fe^3+^ + •OH + OH^−^) under the influence of iron overload is considered the main cause of such toxicity in the body. The high reactivity and low specificity of •OH makes it extremely difficult to defend cell components and body fluids against this action. Under oxidative stress, the activation of the immune system and the stimulation of inflammatory cytokines (e.g., IL-6) are also observed, which are early defensive reactions of the body. This is perceivable of how oxidative stress is “maintained” during post-exercise restitution. The distortion of physiological homeostasis and increase of ionized iron concentration in the bloodstream due to intense exercise, which is a consequence of this process, on the one hand, may contribute to the intensification of free radical reactions [4] and contribute to the increase in inflammation and disturbances in iron metabolism, on the other hand. Research shows that female athletes, compared to their male peers who do not practice sports, are more likely to suffer from disturbances in iron homeostasis in their bodies [5,6]. Moreover, the post-exercise changes in parameters related to iron metabolism may result in an increased frequency of pathogenic infections in female athletes and a decrease in their ability to perform intense exercise [7].

The literature analysis confirms that the bioactive compounds present in bovine colostrum (BC), particularly enzymatic and non-enzymatic antioxidants [8,9], may result in mitigating the adverse effects of oxidative stress. In addition, the presence of lactoferrin, casein, and whey proteins in BC may enhance this positive interactive effect through their ability to chelate iron ions [10].

Therefore, the usage of BC dietary supplement with strong antioxidant and anti-inflammatory properties can be a clinically relevant, relatively effective, and safe intervention, not only for protecting the body against oxidative stress induced by intense physical exercise but also for reducing inflammation and simultaneously having a positive mitigation effect on the iron metabolism parameters. Unfortunately, few studies published have analyzed the effectiveness of BC as a dietary supplement that may be involved in the regulation of antioxidant capacity. Animal model studies by Appukuta et al. [11] showed the beneficial effects of BC in mice, which were manifested by the reduction of oxidative damage in skeletal muscles induced by physical exercise.

This clinical trial was designed to evaluate the impact of a long-term (six months supplementation) of bovine colostrum in female basketball players on iron homeostasis as well as a potentially effective and safe therapy to reduce the harmful effects of intense physical activity on the body of competitive players.

## 2. Materials and Methods

### 2.1. Participants

The research was conducted from April to October 2018 on female basketball players of the youth groups of ENEA AZS AJP Gorzów Wlkp. (1st and extra class basketball league). The first test date (April 2018) was at the end of the starting period (the players finished the games). The second test (October 2018) fell during the period of immediate preparation for competition, characterized by the highest effort load for the competitors. The participant’s characteristics are presented in Table 1.

Experimental procedures and potential risks were discussed with the participants, and informed consent forms were provided and signed prior to inclusion in the study. The study was conducted in accordance with the Declaration of Helsinki, and its protocol was approved by the local Ethics Committee at Poznan University of Medical Sciences (Decision no. 714/18 of 14 June 2017).

### 2.2. Diet Supplementation

Before supplementation, the players were randomly divided into two groups (Figure 1). The supplemented group (n = 11) received four capsules of BC (produced by AGRAPAK, Poland) every morning and evening. One gel capsule contained 0.4 g of colostrum. The composition of the supplement per single dose of 3.2 g (four capsules) was: total protein—2.620 g, lactose—0.16 g, fat—0.05 g, and active protein substances (lactoferrin—30 mg, PRP—0.16 g, IgG—1050 mg, IGF—16 µg, LZM—21.2 mg, and αLA—30 mg). The PRP content was estimated by measuring the content and ratio of amino acids (Pro and Val) based on the conducted research and analysis of bibliographic data [12]. The analysis of the content of IgG, lactoferrin, lactalbumin, and lysozyme in the gastro-intestinal content after simulating the digestion of the encapsulated Colostrum PRP preparation did not change and was similar to the content of these compounds in the undigested preparation. Lactoferrin level analysis was performed using the Bovine LTF ELISA Kit (SunRed Biological Technology Co., Ltd., Shanghai China). The placebo group (n = 9) was given the same doses of powered milk in similar capsules. The composition of the placebo capsules for a single dose of 3.2 g consisted of lactose 1.6 g, protein 1.08 g, fat 0.04 g, and ash 0.25 g. The treatment period lasted for 24 weeks in total.

### 2.3. Physical Performance Examination

Before and six months after supplementation, all players performed the maximum stress test on the HP Cosmos treadmill (Germany). During the test, the aerobic capacity of the participants was assessed. The test protocol was as follows: the starting speed of the treadmill for the runners was 8.0 km/h, then it was increased every 2 min by 1.0 km/h until exhaustion. Participants were verbally encouraged to continue for as long as possible (Figure 2). 

Heart rate (bpm) was recorded with a sport tester (Polar PE 3000). The data are presented in Table 2.

### 2.4. Data Collection and Examination

At each of the time points, before the exercise, in the first minute after the end of testing, and after 3 h of restitution, blood was collected from the participants from the antecubital vein. Blood samples were collected in tubes with dipotassium ethylene diamine tetra-acetic acid (K2EDTA) as an anticoagulant. The list of analyzed parameters included: red blood cell (RBC), hemoglobin (Hb), and hematocrit (HTC), all determined with the MYTHIC 18 Hematology Analyzer (Orphee Medical, Geneva, Switzerland). Relative changes in plasma volume were calculated from blood hematocrit and hemoglobin concentrations using Dill and Costill’s equation [13].

All assays were performed using the Enzyme-Linked Immunosorbent Assay (ELISA). All kits were used in accordance with the manufacturer`s instructions. The following ELISA kits were used for a detailed analysis of changes in the levels of SOD (SOD Activity Assay Kit, SunRed): TBARS (Human Thiobarbituric Acid Reactive Substance ELISA Kit; SunRed), PC (Human Protein Carbonyl ELISA Kit; SunRed), TAC (Total Antioxidant Capacity ELISA Kit; Omnignostica Forschungs GmbH), IL-6 (Human Interleukin 6 ELISA Kit; SunRed), Hepcidin 25 (DRG Hepcidin 25 ELISA Kit), ferritin (Human Ferritin ELISA Kit; DRG), transferrin (Transferrin Human ELISA Kit; SunRed), HPX (Human Hemopexin ELISA Kit; SunRed), Lactoferrin (Human Lactoferrin AssayMax ELISA Kit; AssayPro), and myoglobin (Myoglobin ELISA-DRG). Iron concentration and TIBC were measured using the colorimetric method (BioMaxima, Poland). The unsaturated iron-binding capacity (UIBC) was calculated from the formula: UIBC = TIBC − Fe. Transferrin saturation was calculated as serum iron/TIBC. The Thermo Scientific Multiscan GO Microplate Spectrophotometer produced by Fisher Scientific Finland was used for the material examination.

### 2.5. Statistical Analysis

Statistical analyses were performed using STATISTICA 13.0 (StatSoft, Krakow, Poland), and a graphical representation was prepared using GraphPad Software (San Diego, CA, USA). The normality and homogeneity of data variables were checked using the Shapiro-Wilk and Levene’s tests, respectively. Following normality testing, variables were analyzed using a two-way mixed model analysis of variance (ANOVA), with the group as the between-group factor (PLACEBO and COLOSTRUM) and time as a within-group factor (Pre, Post, Recovery). Significant interactions and main effects were analyzed using Tukey’s post hoc test. The results achieved during the exercise test were subjected to a one-way analysis of variance (ANOVA). Effect size calculations were performed using Cohen’s D for pairwise comparisons, with values >0.8, 0.5–0.8, 0.2–0.5, and <0.2 considered as large, moderate, small, and trivial, respectively. Associations among measured parameters were analyzed using Pearson’s linear regression (coefficient, r). For all analyses, significance was set at *p* < 0.05. Data are represented as a means ± SD.

## 3. Results

### 3.1. Oxidative Stress

The results presented in Table 2 show that no significant changes occurred in the exercise parameters measured during the exercise test performed before the start of the study, as well as no marked differences were observed in the tested athletes after six months of supplementation. 

The noteworthy influence of the exercise test was observed on TAC levels. In the first stage of the study, in the placebo group, a significant increase in TAC concentration was observed after the exercise test (*p* < 0.05 Pre vs. Recovery and Post vs. Recovery) and in the second group (colostrum) (*p* < 0.01 Pre vs. Post and Pre vs. Recovery). In the second stage of the study, after six months of supplementation, no differences were found in the results obtained for either group (Figure 3).

Before supplementation, there was a statistically significant increase in SOD concentration in the placebo group after exercise (*p* < 0.05 Pre vs. Post), while during the restitution period, the value decreased to the baseline level (*p* < 0.05 Post vs. Recovery). In the second stage of the study, there were no significant changes in SOD concentration in either group; however, pairwise comparisons showed a significant difference between SOD placebo and SOD colostrum during recovery, and in the colostrum group, this value was significantly higher (*p* < 0.01) (Figure 3). Additionally, large size effects were observed between groups at all timepoints.

There were no significant changes in the concentration of PC in both groups and in both study periods. Additionally, the concentration of TBARS did not change significantly in the first period of the study, before the supplementation. Afterwards, the level of TBARS remained on the same level in placebo group, while after colostrum treatment it was significantly increased (Pre vs. Recovery, *p* < 0.01 and Post vs. Recovery, *p* < 0.01). The pairwise comparison showed significantly higher TBARS levels in the placebo group (*p* < 0.01) for all pairs in the study periods compared to colostrum (Figure 3). The large size effect was observed between groups in all time points. 

### 3.2. IL-6 and Iron Homeostasis

In the first stage of the study, an increase in iron concentration was observed in the placebo group, while a significant decrease was observed during the restitution period (*p* < 0.01 Post vs. Recovery). In the second stage of the study, no significant changes in iron concentration were observed, neither in the study groups nor in pairs in the individual study periods (Figure 4).

In the first stage of the study, the concentration of IL-6 increased significantly both in the placebo group (*p* < 0.05 Pre vs. Post) and the colostrum group (*p* < 0.05 Pre vs. Post). In the second stage, an increase in IL-6 concentration was observed in both groups; however, a significant increase in IL-6 concentration was observed only in the colostrum group (*p* < 0.05 Pre vs. Post). Pairwise comparisons showed that the resting values of the IL-6 level were higher (*p* < 0.01) in the placebo group compared to the same time point in the colostrum group (Figure 4).

In the first stage of the study, a significant decrease in the concentration of lactoferrin and maintenance of low values during the restitution period after the exercise test were observed in both the placebo (*p* < 0.001 Pre vs. Post and Pre vs. Recovery) and colostrum (*p* < 0.001 Pre vs. Post and Pre vs. Recovery) groups. While in the second stage of the study, in the placebo group, the level of lactoferrin decreased slightly after exercise, while in the colostrum group, the level of lactoferrin increased after exercise (*p* < 0.01 Pre vs. Post) and returned to baseline during the restitution period (*p* < 0.01 Post vs. Restitution). Assessment of pairwise comparisons showed significant differences in post-exercise values in the study groups (*p* < 0.05) after supplementation (Figure 4).

In the first stage of the study, there were no significant changes in transferrin concentration in either study group. In the second stage of the study, in the placebo group, a decrease in transferrin concentration during the restitution period was found (*p* < 0.05 Post vs. Restitution). While in the supplemented group, an increase in transferrin concentration was observed during the restitution period (*p* < 0.05 Post vs. Recovery). The pairwise comparison also showed a significantly higher level of transferrin in the recovery time point (*p* < 0.01) in the colostrum group compared to the placebo group (Figure 4). 

In the second stage of the study, there was also a significant negative correlation between the concentration of TBARS and iron in the colostrum group after supplementation (r = −0.4267; *p* = 0.0452); at the same time, no correlation was found in the placebo group (Figure 5).

Among the parameters related to the characteristics of iron homeostasis, no significant changes in hepcidin concentration were observed in the first stage of the study in either study group; while in the second stage of the study, hepcidin levels increased significantly in the placebo group (*p* < 0.05 Pre vs. Post and Pre vs. Recovery). In the colostrum group, a significant increase was observed only after exercise (*p* < 0.05 Pre vs. Post) (Table 3). 

No significant changes were observed in the level of ferritin in either stage of the study. In the case of hemopexin, in the first stage of the study, no significant changes were observed in either group, while in the second stage, a significant increase of ferritin was observed after exercise in the supplemented group (*p* < 0.05 Pre vs. Post and Pre vs. Recovery) (Table 3). 

### 3.3. RBC, Hb, HTC, and Myoglobin

Changes in the number of red blood cells during the study were characterized by a significant decrease in the first stage of the study during the restitution period (*p* < 0.05 Post vs. Recovery in placebo and *p* < 0.05 Post vs. Recovery in colostrum). In the second stage of the study, a decrease was observed only in the placebo group (*p* < 0.05 Post vs. Recovery), while in the supplemented group, the number of red blood cells did not change significantly. Additionally, in the second stage of the study, pairwise comparisons showed a significant difference in the number of red blood cells between the placebo and colostrum groups at the post-exercise time point (Table 4). In the first stage of the study, the concentration of myoglobin significantly increased in both study groups (*p* < 0.05 Pre vs. Post and Pre vs. Recovery, respectively). In the second stage of the study, no significant changes in myoglobin concentration were observed in either group (Table 4).

## 4. Discussion

Physical training is a key factor in enhancing the antioxidant potential of the body, protecting the athlete from oxidative stress induced by exercise [14]. However, literature analysis shows that a stable redox balance concerns efforts of moderate intensity and duration [15,16]. On the other hand, an increase in exercise intensity to maximal and submaximal values, a substantial increase in the duration of exercise, and additionally the accumulation of significant exercise loads may increase the production of free radicals, leading to an increase in the peroxidation modification of lipids, proteins, or nucleic acids [17,18]. Increased intensity of exercise may also be associated with the gradual depletion of the antioxidant protective barrier, which is reflected in the decrease in the concentration of intracellular antioxidant enzymes and glutathione as well as extracellular non-enzymatic antioxidants [19].

Post-exercise intensification of free radical reactions may indicate the insufficiency of the endogenous antioxidant defense mechanism of the body. In competitive athletes, not all training periods are associated with the intensification of these changes. The literature data indicate that during periods of intensive preparation for competitions and periods related to participation in competitions, it is necessary to strengthen the endogenous defense of the body [20,21]. The above thesis is also confirmed by the results of the tests conducted in this study on basketball players. The first stage of the study was conducted during the preparatory period, while the second part was at the final stage of the starting period. The group receiving a placebo showed a significantly higher level of TBARS compared to the group supplemented with BC (Figure 3). The key role in limiting lipid peroxidation in basketball players was played by lactoferrin (LTF) present in BC. Kruzel et al. [22] presented a diagram of the action of lactoferrin, which in the first stage inhibited the reactivity of free ferric ions (Fe + 3), thus limiting the formation of ferrous ions (Fe + 2). The final result was a reduction in the production of hydroxyl radicals (**·**OH) and lipid peroxidation. Studies conducted on mice that performed 30 min of daily treadmill exercise showed that BC supplementation reduced lipid hydroperoxide and xanthine oxidase levels and increased SOD and total antioxidant levels measured in the leg muscle tissue of these mice [11]. It should be emphasized that exercise-induced lipid peroxidation of muscle cell membranes results not only in a decrease in fluidity but also in an increase in their non-specific permeability and the development of inflammation [23].

The analysis of IL-6 changes showed that the six month supplementation of BC athletes produced less of this cytokine at rest compared to the control group (Figure 4). Comparable results were demonstrated by Deros et al. [24], who supplemented adult, type II diabetes patients via a whey protein isolate with a high content of native cysteine and standardized content of lactoferrin. After three months, the supplementation group showed improvement in blood lipid parameters, a decrease in the level of the inflammatory markers TNFα and IL-6, and improvement in the parameters of prooxidative-antioxidant balance, expressed by an increase in the activity of SOD, GPX, and GSH and a reduction in the level of lipid peroxidation (MDA).

The BC -supplemented group also showed a negative correlation between the TBARS level and the iron level (Figure 5). The source of available iron after exercise may be both RBC hemolysis and ROS contributing to the release of iron ions from ferritin. In the present study, we did not observe significant alterations in ferritin levels (Table 3), and therefore, it can be assumed that hemolysis of RBCs occurred due to the negative effect of TBARS on cellular structures, especially on the cell membrane’s condition. The reduction of lipid peroxidation in our study may show a protective effect on red blood cell membranes, which was relatively constant in the group supplemented with RBC. In contrast, the placebo group had a significant decrease in the number of red blood cells, which may indicate hemolysis, leading to an increase in iron concentration in the blood (Table 4). This thesis is also confirmed by other research results [25,26], which showed that the deformability of RBC membranes might decrease after exercise, which leads to increased hemolysis. ANOVA also showed no effect of BC supplementation on hemopexin levels. However, it is worth mentioning that statistically significant changes in the level of this parameter occurred only in the group of athletes supplemented with BC. Higher levels compared to the resting values were noted both in the study immediately after exercise and after 3 h of restitution (Table 4). Hemopexin is one of the antioxidants whose role is to bind free heme, which is formed as a result of increased hemolysis, and thus protects the tissue against oxidative damage [27].

As already mentioned, lactoferrin binds Fe^3+^ ions with high affinity and maintains them in an acidic environment, which is of key importance during intense physical exercise where the body is significantly acidified. There are also other siderophilins in the body, such as transferrin, which also has the ability to reversibly bind Fe^3+^ ions. The level of transferrin in the serum of the tested basketball players, after 3 h of rest, increased in the supplemented group, while in the control group, a decrease in this parameter was observed (statistically significant difference between the groups) (Figure 4). A markedly significant increase in transferrin concentration in the supplemented group may indicate depletion of iron stores in the body 3 h after exercise. This is confirmed by the changes in UIBC, which tend to increase the share of the unsaturated part of transferrin. On the other hand, in the placebo group, transferrin is significantly reduced, which may indicate a persistently higher concentration of iron in the circulating blood as a result of physical exercise. This is reflected in the UIBC values, which show a downward trend, indicating a lower amount of iron needed for complete transferrin saturation (Table 3). The demonstrated changes in transferrin levels also indicate a reduction in inflammation in the BC-supplemented group. Reinke et al. [28] analyzed changes in the parameters of iron metabolism in competitive athletes (footballers and rowers). The research was conducted in three training periods: competition, recovery, and preparatory stages. In both groups, a significantly lower level of transferrin was demonstrated in the starting period compared to the recovery period. These authors also showed unfavorable changes in the parameters of iron metabolism in the tested athletes, which indicated that at the end of the starting period as many as 27% of all athletes had absolute iron deficiency.

In other studies [29], the cumulative effect of training stress in highly qualified rowing athletes on hepcidin levels and its impact on parameters related to iron metabolism was analyzed. The cited authors showed that the level of hepcidin and ferritin as acute phase proteins was a sensitive indicator of changes in training loads (increase in exercise volume and intensity). The discovery of the hormone hepcidin has been linked to inflammation and distortion of iron metabolism in the body. The results of the studies published so far [30,31] indicate that inflammation contributes to an increased level of hepcidin, which through the degradation of ferroportin, not only reduces the absorption of iron from the intestine but also reduces its release from stores (spleen, liver). If inflammation persists in the body for a longer period, it may result in the development of hypochromia and then anemia. In both groups of basketball players in the second period of the study (after the supplementation), the level of hepcidin after exercise increased significantly compared to the value prior to exercise. After 3 h of rest, the elevated level of this parameter was maintained only in the control group (Table 3). However, the effect of BC supplementation on Fe levels has not been demonstrated (Figure 4).

## 5. Conclusions

This study conducted on a group of basketball players who received a long-term bovine colostrum supplementation resulted in a reduction in the harmful effects of free radicals, which manifested in a significantly lower level of lipid peroxidation. The reduction of oxidative stress and inflammation, along with the alleviation of the inflammatory response, also had a positive effect on the stabilization of iron metabolism parameters. The obtained research results indicate that BC may not only be significantly helpful in planning the diet of athletes but BC supplementation can be considered a potential natural chelator of iron ions.

## Figures and Tables

**Figure 1 nutrients-15-00186-f001:**
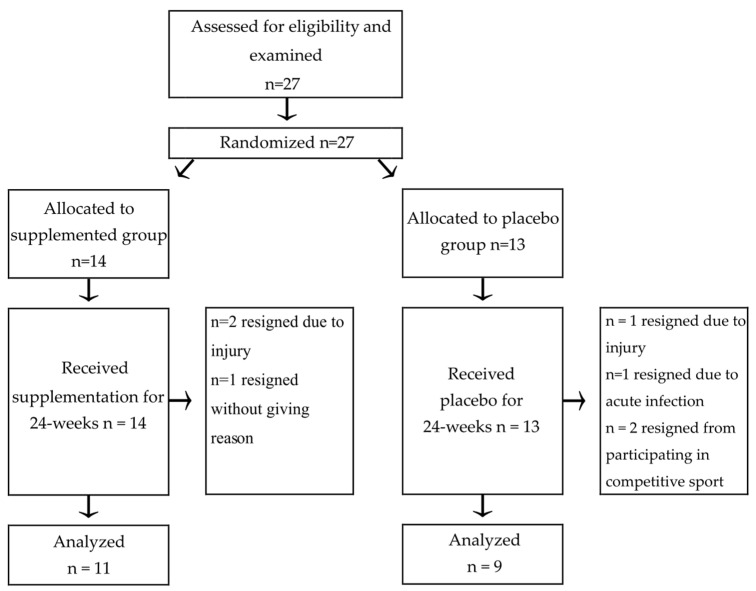
The flow chart shows the recruitment process of participants in the trial.

**Figure 2 nutrients-15-00186-f002:**
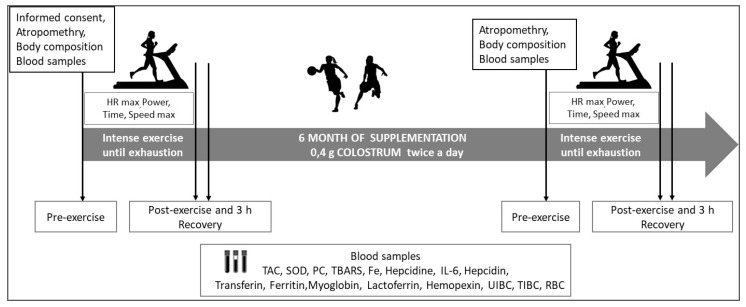
Study design.

**Figure 3 nutrients-15-00186-f003:**
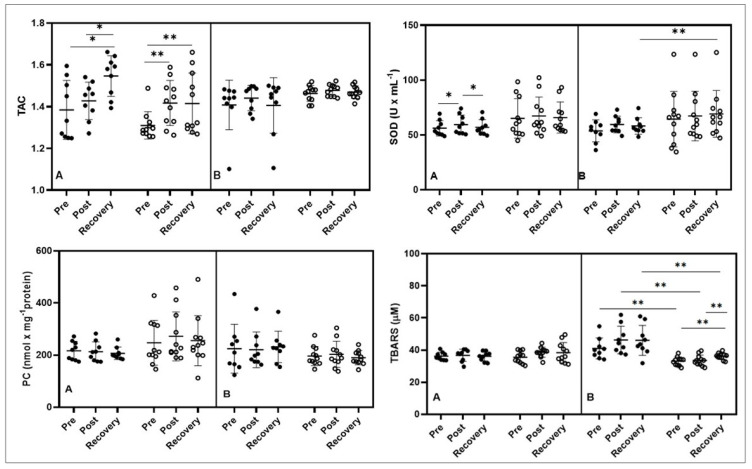
Effect of bovine colostrum supplementation on TAC—total antioxidant capacity; SOD—superoxide dismutase; PC—protein carbonyl; and TBARS—thiobarbituric acid reactive substances during exercise test performed before (**A**) and after supplementation (**B**). Note. Values are presented as mean ± SD; PLACEBO (black dots); COLOSTRUM group (white dots); * *p* < 0.05 ** *p* < 0.01; TAC total antioxidant capacity; SOD superoxide dismutase; PC protein carbonyl; TBARS thiobarbituric acid reactive substances.

**Figure 4 nutrients-15-00186-f004:**
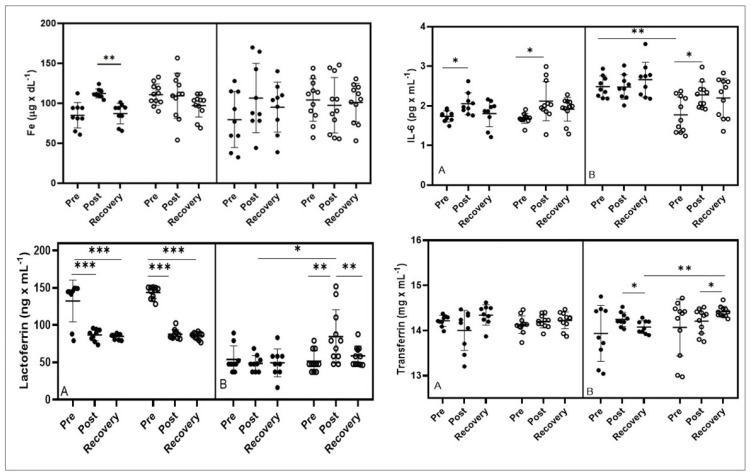
Effect of bovine colostrum supplementation Fe-iron; IL-6-interleukin 6, lactoferrin, and transferrin concentrations during exercise test performed before (**A**) and after supplementation (**B**). Note. Values are presented as mean ± SD; placebo (black dots); colostrum group (white dots); * *p* < 0.05 ** *p* < 0.01 *** *p* < 0.001.

**Figure 5 nutrients-15-00186-f005:**
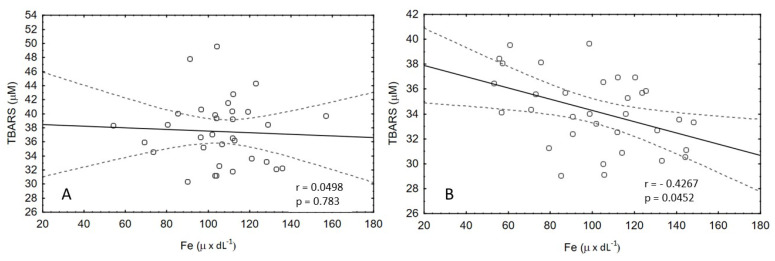
Correlation between TBARS and iron concentrations in colostrum before (**A**) and after supplementation (**B**).

**Table 1 nutrients-15-00186-t001:** Basic characteristics of the studied groups (mean ± standard deviation).

Parameters	BC Supplemented Group (n = 11)	Control Group (n = 9)
Age [years]	17.09 ± 1.24	16.0 ± 0.67
Body mass [kg] Body height [cm]	67.2 ± 6.66 177.4 ± 5.91	65.6 ± 7.02 169.5 ± 4.36

**Table 2 nutrients-15-00186-t002:** Basic characteristics of the studied groups before and after supplementation (mean ± standard deviation (SD)).

Parameter	Group	Time Point I (mean ± SD)	Time Point II (mean ± SD)
HR max (bpm)	Control	192.13 (12.29)	190.9 (8.76)
	Supplemented	189.0 (18.46)	188.5 (7.32)
Power (watt)	Control	166.3 (21.29)	162.3 (18.94)
	Supplemented	187.4 (17.88)	187.4 (18.97)
Watt/kg	Control	2.51 (0.17)	2.46 (0.16)
	Supplemented	2.78 (0.21)	2.76 (0.15)
Time (s)	Control	643.33 (96.18)	596.67 (125.0)
	Supplemented	781.82 (126.63)	774.55 (88.81)
Speed max (km/h)	Control	12.6 (0.87)	12.3 (0.80)
	Supplemented	13.9 (1.11)	13.78 (0.84)

Note: HRmax—maximal heart rate.

**Table 3 nutrients-15-00186-t003:** Concentrations of Hepcidin, Hemopexin, Ferritin, UIBC, and TIBC of athletes supplemented with colostrum bovine and with placebo.

Variables	Before Supplementation			*p* Value			After Supplementation			*p* Value		
	Pre-Exercise x ± SD	Post-Exercise x ± SD	Recovery x ± SD	Pre CON vs. SUPL	Post CON vs. SUPL	Recovery CON vs. SUPL	Pre-Exercise x ± SD	Post-Exercise x ± SD	Recovery x ± SD	Pre CON vs. SUPL	Post CON vs. SUPL	Recovery CON vs. SUPL
**Hepcidin** (ng × dL^−1^)												
PLACEBO	15.91 ± 1.29	14.81 ± 1.95	15.25 ± 2.70	0.213	0.890	0.979	19.10 ± 3.54	22.83 ± 2.60 ^a^	22.50 ± 3.00 ^b^	0.460	0.402	0.149
COLOSTRUM	14.91 ± 2.29	14.55 ± 1.82	14.89 ± 2.04				17.27 ± 3.83	21.19 ± 2.75 ^a^	18.98 ± 3.33			
**Hemopexin** (µg × mL^−1^)												
PLACEBO	614.64 ± 156.38	607.97 ± 138.93	672.14 ± 181.18	0.524	0.773	0.750	613.00 ± 108.27	685.78 ± 150.34	600.67 ± 179.83	0.722	0.551	0.894
COLOSTRUM	653.02 ± 144.18	614.61 ± 126.08	676.20 ± 150.73				616.33 ± 140.10	714.00 ± 152.41 ^a^	655.85 ± 114.07 ^b^			
**Ferritin** (ng × mL^−1^)												
PLACEBO	88.55 ± 11.95	91.84 ± 15.84	86.43 ± 10.81	0.732	0.508	0.802	99.71 ± 43.50	90.84 ± 9.13	89.27 ± 6.96	0.638	0.431	0.313
COLOSTRUM	92.51 ± 10.47	90.08 ± 11.97	89.67 ± 10.74				93.84 ± 11.07	89.50 ± 8.50	86.92 ± 8.02			
**UIBC** (µg × dL^−1^)												
PLACEBO	239.71 ± 86.10	259.76 ± 86.33	182.27 ± 49.26 ^b^	0.777	0.934	0.525	255.52 ± 112.27	249.84 ± 71.29	200.45 ± 69.98	0.631	0.909	**0.054**
COLOSTRUM	224.33 ± 55.50	228.22 ± 93.67	245.15 ± 85.92				218.29 ± 58.55	245.97 ± 52.77	261.89 ± 55.26			
**TIBC** (µg × dL^−1^)												
PLACEBO	325.59 ± 59.78	374.75 ± 66.40 ^a^	297.77 ± 57.89 ^b^	0.614	0.658	0.254	332.98 ± 83.48	367.55 ± 112.86	317.85 ± 83.98	0.995	0.582	0.327
COLOSTRUM	337.68 ± 50.22	360.19 ± 116.69	346.94 ± 83.82				322.51 ± 55.26	332.25 ± 65.02	353.76 ± 51.53			

Note. Values are presented as mean ± SD; Significant differences *p* < 0.05; ^a^ Pre-exercise vs. Post-exercise; ^b^ Pre-exercise vs. Recovery. TIBC, total iron-binding capacity; UIBC, unsaturated iron-binding capacity. Bold values denote statistical significance at the *p* < 0.05 level.

**Table 4 nutrients-15-00186-t004:** Concentrations of RBC, Hb, HTC, and Myoglobin of athletes supplemented with colostrum bovine and with placebo.

Variables	Before Supplementation			*p* Value			After Supplementation			*p* Value		
	Pre-Exercise x ± SD	Post-Exercise x ± SD	Recovery x ± SD	Pre CON vs. SUPL	Post CON vs. SUPL	Recovery CON vs. SUPL	Pre-Exercise x ± SD	Post-Exercise x ± SD	Recovery x ± SD	Pre CON vs. SUPL	Post CON vs. SUPL	Recovery CON vs. SUPL
**RBC** (10^9^ × mL^−1^)												
PLACEBO	4.36 ± 0.37	4.52 ± 0.37	4.19 ± 0.34 ^c^	0.929	0.936	0.094	4.63 ± 0.45	4.88 ± 0.45	4.31 ± 0.27 ^c^	0.998	**0.019**	0.873
COLOSTRUM	4.22 ± 0.25	4.35 ± 0.27	3.83 ± 0.37 ^b^				4.50 ± 0.45	4.35 ± 0.27	4.42 ± 0.21			
**Hb** (g × L^−1^)												
PLACEBO	13.51 ± 1.20	13.89 ± 1.13 ^a^	12.62 ± 1.16 ^b^	0.962	0.994	0.540	13.93 ± 1.07	14.80 ± 1.75	12.95 ± 0.78 ^b^	0.845	0.579	0.999
COLOSTRUM	13.21 ± 0.78	13.54 ± 0.75 ^a^	11.95 ± 0.91 ^b^				13.27 ± 0.44	13.99 ± 0.73	12.80 ± 1.01 ^b^			
**HTC** (%)												
PLACEBO	37.84 ± 3.05	39.13 ± 2.93 ^a^	35.82 ± 2.83 ^b^	0.823	0.788	**0.023**	40.95 ± 3.27	43.73 ± 4.86	37.94 ± 2.30 ^b^	0.772	0.376	0.999
COLOSTRUM	36.74 ± 2.01	37.6 ± 1.87 ^a^	32.76 ± 2.57 ^b^				38.89 ± 1.35	41.01 ± 1.87 ^a^	37.70 ± 2.67 ^b^			
**Mioglobin** (µg × L^−1^)												
PLACEBO	17.11 ± 5.03	19.06 ± 3.04 ^a^	29.69 ± 3.85 ^b^	0.309	0.203	0.994	25.69 ± 8.31	27.27 ± 6.60	29.60 ± 7.25	0.964	0.671	0.716
COLOSTRUM	19.27 ± 3.50	21.21 ± 2.13 ^a^	29.41 ± 2.92 ^b^				26.51 ± 10.87	28.25 ± 8.43	29.82 ± 7.47			

Note. Values are presented as mean ± SD; Significant differences *p* < 0,05; ^a^ Pre-exercise vs. Post-exercise; ^b^ Pre-exercise vs. Recovery; ^c^ Post-exercise vs. Recovery. RBC red blood cells; Hb hemoglobin; HTC hematocrit; TIBC, total iron-binding capacity; UIBC, unsaturated iron-binding capacity. Bold values denote statistical significance at the *p* < 0.05 level.

## Data Availability

Due to ethical concerns, the datasets generated and/or analyzed during the current study and supporting data cannot be made openly available; however, they are available from the corresponding author upon reasonable request.

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
