# Peer review of "Effects of Long-Term Supplementation of Bovine Colostrum on Iron Homeostasis, Oxidative Stress, and Inflammation in Female Athletes: A Placebo-Controlled Clinical Trial"

_nutrients, 2022, doi:10.3390/nu15010186_

Round 1
Reviewer 1 Report
It is necessary to separate the results by type and explain them. Such as 3-1 Exercise parameters.
Author Response
Thank you very much for the comments on our paper entitled „Effects of long-term supplementation of bovine colostrum on iron homeostasis, oxidative stress, and inflammation in female athletes”. The manuscript has been revised following all the remarks of the expert referees.

Reviewer 2 Report
Dear Authors,
The manuscript presents a theme of significant scientific relevance. However, I have some comments.
- I recommend an English review of the manuscript. For example, in line 28, "harmful effects prodiced by the free radicals" I suggest "harmful effects produced by the free radicals"
- What is the methodology for TBARS and SOD activity?
- TBARS is not a significant experiment for lipid peroxidation. I suggest 4-HNE.
- What is the reason for measuring the antioxidant? How about catalase and glutathione peroxidase?
- How the authors analyzed the decrease in free radicals?
Author Response

(The authors gave the same response as above.)

Reviewer 3 Report
This manuscript described the effects of long-term supplementation of bovine colostrum on iron homeostasis, oxidative stress, and inflammation in female athletes. The topic is interesting, however, it has not been ready for publication in the present form.
1.The English writing should be polished by a native speaker, there were many errors in the context, such as Lines 17-18 and 25-26…
2.The abbreviation definition was not correct, such as free oxygen radicals (ROS).
3.The biochemical testing methods should be more detailed (section2.4).
4. In the INTRODUCTION and DISCUSSION sections, the authors emphasized the importance of iron balance (Lines 46-53, Lines 355-357). However, the results of this study showed that changes in iron concentration were observed neither in the study groups nor in pairs in the individual study periods (Lines 186-188). In this case, how did the authors reach such a conclusion “In consequence, alleviation of the inflammatory response may also cause positive action on iron homeostasis (Lines 28-29).
5. The previous study showed that the increase of ionized iron concentration in the blood serum, which is a consequence of this process, may contribute to the intensification of free radical reactions. I suggest the authors to examine the indicator “ionized iron concentration”.
Author Response

(The authors gave the same response as above.)

Round 2
Reviewer 2 Report
Dear Authors,
Thank you for your response.
Reviewer 3 Report
I read the authors' responses. Thanks for the revised version.